# Cardiac Imaging Biomarkers in Chronic Kidney Disease

**DOI:** 10.3390/biom13050773

**Published:** 2023-04-29

**Authors:** Silvia C. Valbuena-López, Giovanni Camastra, Luca Cacciotti, Eike Nagel, Valentina O. Puntmann, Luca Arcari

**Affiliations:** 1Department of Cardiology, University Hospital La Paz, 28046 Madrid, Spain; 2Cardiology Unit, Madre Giuseppina Vannini Hospital, 00177 Rome, Italy; 3Institute for Experimental and Translational Cardiovascular Imaging, University Hospital Frankfurt, Theodor-Stern-Kai 7, 60590 Frankfurt am Main, Germany

**Keywords:** Chronic kidney disease, uremic cardiomyopathy, T1 mapping, T2 mapping, cardiac magnetic resonance, myocardial fibrosis

## Abstract

Uremic cardiomyopathy (UC), the peculiar cardiac remodeling secondary to the systemic effects of renal dysfunction, is characterized by left ventricular (LV) diffuse fibrosis with hypertrophy (LVH) and stiffness and the development of heart failure and increased rates of cardiovascular mortality. Several imaging modalities can be used to obtain a non-invasive assessment of UC by different imaging biomarkers, which is the focus of the present review. Echocardiography has been largely employed in recent decades, especially for the determination of LVH by 2-dimensional imaging and diastolic dysfunction by pulsed-wave and tissue Doppler, where it retains a robust prognostic value; more recent techniques include parametric assessment of cardiac deformation by speckle tracking echocardiography and the use of 3D-imaging. Cardiac magnetic resonance (CMR) imaging allows a more accurate assessment of cardiac dimensions, including the right heart, and deformation by feature-tracking imaging; however, the most evident added value of CMR remains tissue characterization. T1 mapping demonstrated diffuse fibrosis in CKD patients, increasing with the worsening of renal disease and evident even in early stages of the disease, with few, but emerging, prognostic data. Some studies using T2 mapping highlighted the presence of subtle, diffuse myocardial edema. Finally, computed tomography, though rarely used to specifically assess UC, might provide incidental findings carrying prognostic relevance, including information on cardiac and vascular calcification. In summary, non-invasive cardiovascular imaging provides a wealth of imaging biomarkers for the characterization and risk-stratification of UC; integrating results from different imaging techniques can aid a better understanding of the physiopathology of UC and improve the clinical management of patients with CKD.

## 1. Introduction

In patients with chronic kidney disease (CKD), an increased rate of adverse cardiovascular (CV) events with a worsening of renal function has been observed [1]. Accordingly, the European Society of Cardiology identifies the presence of renal insufficiency as a marker of increased risk of coronary artery disease (CAD), which is higher as renal dysfunction worsens [2]. However, despite this well-defined association with CAD, CV events in patients with CKD are largely driven by non-atherosclerotic pathologies, especially at higher degrees of renal dysfunction [3]. Indeed, the causes of death in patients with end-stage renal disease (ESRD) are largely attributable to heart failure, often with preserved ejection fraction (HFpEF), and related sudden cardiac death [4]. A major correlate and determinant of these outcomes is the pathologic cardiac remodelling caused by renal dysfunction, which is termed “uremic cardiomyopathy” (UC) [5]. Its underlying pathophysiology is rather complex, extending beyond the clustering of traditional risk factors such as diabetes and hypertension, and relies on effects resulting from pressure and volume overload as well as CKD-related factors [4]. These factors include, among others, inflammation, anemia, oxidative damage and disruption of bone metabolism [6,7], which together contribute to the peculiar myocardial changes detected in UC. Briefly (Table 1), UC is characterized by cardiomyocyte hypertrophy and interstitial expansion due to diffuse fibrosis, with subtle myocardial edema and replacement fibrosis that can also be present. Morphology of the cardiac chambers is characterized by left ventricular hypertrophy (LVH) and right ventricular (RV) dilation. Diastolic dysfunction and, later, systolic dysfunction can develop. Vascular involvement is mainly characterized by the presence of vascular stiffness and calcification. Notably, these changes at the CV level constitute markers of progressive disease and worse prognosis. Different CV imaging modalities can provide non-invasive detection and quantification of UC-related cardiac and vascular abnormalities, and indeed several image biomarkers with relevant prognostic implications have been identified in this setting (Table 1). The aim of the present review is to summarize the existing evidence on the role of CV imaging in the assessment of CKD-related myocardial remodelling, with a focus on echocardiography, cardiac magnetic resonance (CMR) imaging and computed tomography (CT).

## 2. Echocardiography

### 2.1. Cardiac Remodelling

Echocardiography represents the first-line modality used for the investigation of UC, and many data are currently available from the literature. Though limited by geometric assumptions, especially when 2D imaging is used [66], the evaluation of LV mass and cardiac dimensions represents an important step in the echocardiographic evaluation of UC. The increase in LV mass assessed by 2D-echocardiography is a well-known marker of adverse cardiac remodelling, which is associated with worsening renal function and higher rates of adverse outcomes [8]. The pattern of geometric remodelling has its own implications. Indeed, the progression of UC has been classically described as the development of concentric remodelling, followed by concentric (wall thickness-to-radius ratio > 0.42) and then eccentric hypertrophy with dysfunction. Though this remains a simplistic representation, several longitudinal studies have reported this pathway [67,68]. Consistent with the notion of eccentric hypertrophy as a later stage of UC, it is associated with higher mortality rates [8], especially sudden cardiac death [9]. On the other hand, concentric LVH has been found to increase rates of cerebrovascular events [10]. In this context, the use of 3D-echocardiography might provide more accurate measurements of LVH and LV function, and it has been used in patients with CKD showing progressive adverse remodelling and reduced function with the severity of renal disease [11].

### 2.2. Diastolic Function

Changes in myocardial function characterize the natural history of UC, with diastolic rather than systolic abnormalities representing a key feature. A longitudinal study analysing patients with ESRD undergoing replacement therapy found that diastolic abnormalities preceded the reduction of left ventricular ejection fraction, which became evident only at year three after the baseline evaluation [36]. Furthermore, diastolic dysfunction represents a common finding in CKD and is linked to poor prognosis [37]. Several echocardiographic techniques can be used to assess diastolic function, including pulsed wave, continuous wave and tissue Doppler imaging (Figure 1) [69], albeit retaining poor accuracy compared with the gold-standard of invasive evaluation [70]. Nevertheless, previous studies in CKD patients have shown that an increasing E/e’ ratio correlates with higher rates of cardiovascular events [38], making this simple and easily obtainable marker of diastolic function valuable in this setting. Remodelling of atrial chambers (i.e., increased left atrial volume) is prognostic in CKD, where 3D-echocardiography might add value beyond the standard 2D evaluation [71]. Atrial strain provides a parametric assessment of the atrium, including the reservoir, conduit and pump functions, and represents a novel and attractive ultrasound tool for the evaluation of diastolic dysfunction. Increased LV end-diastolic pressure relates to altered atrial mechanics, and the addition of left atrial strain to improve standard work-up of patients with suspected HFpEF has been hypothesized [72]. In CKD, the assessment of atrial strain provided added diagnostic and prognostic value [39], including association with major adverse cardiovascular events [40].

### 2.3. Systolic Function

Systolic dysfunction, as signified by reduced left ventricular ejection fraction (LVEF), develops later in the course of UC but retains prognostic value and might improve after kidney transplantation [42]. Speckle tracking echocardiography provides a more accurate evaluation of systolic function compared to the standard 2D-examination, making it possible to assess cardiac deformation with the ability to detect subtle degrees of systolic dysfunction. In CKD, even in patients with preserved LV ejection fraction, decreasing global longitudinal strain of the LV was independently associated with adverse outcomes in multiple cohort studies [43,44,45]. The right heart is often involved in UC [48], with abnormalities detectable early in the course of the disease, even before the decline of LV ejection fraction [36]. Right ventricular involvement retains prognostic significance in CKD [49]. Of note, the use of 3D-echocardiography can aid a better visualization of this heart chamber, which is characterized by a less regular shape than the LV [33].

### 2.4. Calcification

Vascular and cardiac calcification may be easily identified but less easily quantified by echocardiography, though some methods have been developed for a quantitative calcium evaluation and retain prognostic significance [73]. In patients with CKD, cardiac calcification, as detected by echocardiography, is common [50] and is associated with cardiovascular disease [51] as well as often involving cardiac valves with prognostic relevance [52,53]. Notably, mitral valve calcification seems to retain higher prognostic value compared to other locations, such as the aortic valve [54,55].

### 2.5. Other Biomarkers

Other features of UC are less effectively imaged by echocardiography. Vascular stiffness is hardly imaged directly by echocardiography, where the gold-standard for the assessment of pulse wave velocity (PWV) is calculation tonometry or through mechanotransducers [74]. However, some observations on the use of echocardiography have been reported [75]. Pulse wave velocity using applanation tonometry is a simple tool that has demonstrated a correlation with prognosis in CKD [45]. Furthermore, tissue Doppler imaging of the aortic wall has been described as a potential tool for the evaluation of arterial stiffness [76]; however, to date no specific study using this approach in CKD patients is available. Fibrosis and edema cannot be reliably imaged by echocardiography. Backscatter analysis is a non-invasive tool that can be used to estimate LV fibrosis by assessment of myocardial reflectivity, with values correlating with echocardiography derived indexes of LV stiffness and diastolic dysfunction [77]. However, very few data are currently available—none specifically in the CKD population. Microcirculation is impaired in CKD, with coronary flow reserve decreasing with the worsening of renal dysfunction [34]. Echocardiography can assess microvascular function by Doppler analysis of the left anterior descending artery during adenosine administration, which, in CKD patients, is often impaired [35] and is associated with the severity of underlying anemia [33].

## 3. Cardiac Magnetic Resonance

### 3.1. Left Ventricular Hypertrophy

Most of the evidence that links CKD and LVH derives from studies performed with echocardiography, though CMR offers undeniable advantages. Indeed, echocardiography systematically overestimates myocardial mass [12] and is subject to higher variability [13], which could, at least partially, account for the sometimes conflicting results found in previous studies [14]; conversely, CMR allows an accurate and reproducible measurement of the LV mass based on a slice-per-slice approach rather than on geometrical assumptions. Significant CMR-measured LVH has been reported in patients undergoing hemodialysis compared to controls [15]; however, less information on the earlier stages of CKD is available. In a recent study that included a broad range of pre-dialysis CKD stages, LV mass did not differ across stages 2–4, but significantly increased in stage 5, suggesting that LVH is a late phenomenon in the natural history of the disease [16]. This could limit the use of LV mass as a surrogate endpoint to monitor the effectiveness of medical or interventional therapies; indeed, a recent study failed to show LVH regression 12 months after a kidney transplant compared with patients continuing in dialysis [46]. Myocardial structural and functional changes do occur in the early stages of CKD, including myocyte hypertrophy, expansion of extracellular space due to fibrosis, edema and increased vascular stiffness. The possible non-reversibility of LVH shifts the focus of attention to different, earlier phenomena, amenable to modification by earlier interventions.

### 3.2. Regional Fibrosis

CMR is the technique of choice for non-invasive detection of fibrosis. Late gadolinium enhancement (LGE) imaging detects areas of dense, replacement fibrosis. This assessment implies the use of gadolinium-based contrast agents (GBCAs), which is controversial in patients with advanced renal disease because of the risk of development of systemic nephrogenic fibrosis. However, the risk is negligible with the widespread use of more stable macrocyclic compounds; accordingly, the most recent consensus documents do not restrict its use in CKD, as long as low risk GBCAs are used [78], which even makes it questionable to screen for renal dysfunction before a CMR examination in the outpatient setting [79]. The prevalence of LGE in CKD is relatively high, with reported rates of 28.4–79% in dialysis patients [17]. Among CKD patients not on replacement therapy, LGE is not so common, but a prevalence between 7 and 35% [18] has been described. Two common patterns have been described in these patients: subendocardial distribution, indicating previously known or silent myocardial infarction (Figure 2), and non-ischaemic scar (including patterns such as midwall and epicardial scar or LGE in right ventricular insertion points), which may be related to confluent areas of dense interstitial fibrosis or to inflammatory processes, although its physiopathology is not completely understood (Figure 3). Among dialysis patients, ischaemic etiology features in roughly half of the patients, being non-ischaemic patterns that are much more frequent in less severe CKD, which is likely to reflect a much higher burden of coronary disease and classical cardiovascular risk factors within the first group. Data on the prognostic relevance of the presence of LGE are scarce, but one recent study including 159 pre-dialysis patients (stages 2–5) found no association of LGE with adverse cardiovascular outcomes after 3.8 years [18]. Some limitations apply when considering the use of LGE as an early marker of uremic cardiomyopathy. Originally conceived for ischaemic cardiomyopathy, this technique relies on the identification of a healthy versus a diseased myocardium, and so is limited in the assessment of diffuse interstitial fibrosis. 

### 3.3. Diffuse Fibrosis and Edema

The assessment of diffuse myocardial fibrosis has gained weight in the last few years, with the use of T1 and T2 parametric mapping sequences. Although T1 mapping is very sensitive to myocardial pathology, it lacks specificity; its increase may be due to fibrosis, but also to edema or infiltration. On the contrary, T2 mapping is specifically increased in the presence of myocardial water, therefore the combination of both of these offers more valuable information. Multiple studies have reported significant differences in T1 and T2 between CKD and subjects with normal renal function [19,20,21,22,23,24,25,26,27,80] (Table 2A). These findings include a wide range of CKD patients, not only those under replacement therapy (hemodialysis or peritoneal dialysis), but also moderately diseased patients with CrCl < 60 mL/min/m^2^. Native T1 emerges as an early marker of cardiac disease in CKD, with increased values independent of the presence of LVH and conventional risk factors [28] and mainly driven by CKD-related factors. The hypothesis that diffuse fibrosis is the main driver of the increase in native T1 is consistent with previous histology studies in CKD [81,82] and the extensive available information of T1 in other cardiomyopathies. However, no histological correlate specifically in CKD is currently available, but there is an ongoing trial addressing this question (NCT03586518). The role of T2 mapping has been less extensively studied, although most studies have shown increased values from the early stages of CKD (Table 1). 

Both T1 and T2 were independently related to biomarkers of myocardial injury (hs-TnT) and B-type natriuretic peptides [25,28], showing a stronger relationship with advancing renal failure, all of which suggests a link between increased myocardial water and ongoing myocardial injury in CKD. In a study comparing CKD patients to healthy controls as well as other hypertrophic disease models, such as hypertensive and hypertrophic cardiomyopathy, native T1 was significantly higher in all patient groups compared to controls. However, T2 was specifically increased in CKD, with a strong relationship between the two of them, suggesting that the increase in T1 in these patients might be driven not only by fibrosis, but also, to a certain extent, by increased myocardial fluid [24]. This question has been addressed by several studies that looked into the acute changes in T1 and T2 immediately before and after hemodialysis [29,30] and demonstrated detectable and significant changes in both parameters following hemodialysis. Despite the uncertain association of these changes with global fluid status [22] (either measured by bioimpedance or change in body weight), the most likely explanation is a reduction in myocardial water content [28,29]. Of note, the detection of these subtle changes in myocardial composition is dependent on the timing of the CMR, the fluid status previous to the HD and the intensity of the therapy, making T1 mapping evaluation a potential surrogate endpoint with which to assess the efficacy of different hemodyalisis schemes [31]. On the contrary, the role of myocardial edema was negligible in a study that failed to show a decrease in native T1 and T2 early after kidney transplantation (8 weeks), supporting the hypothesis that increased T1 is mainly driven by fibrosis in uremic cardiomyopathy [83].

A recent cross-sectional study, including the whole range of renal disease (stages 2–5), demonstrated a stepwise increase in native T1 and T2 and serum biomarkers with every stage of CKD [16]. Moreover, T1 was an independent predictor of peak oxygen uptake during cardiopulmonary exercise testing in this cohort. Although the increase in native T1 and T2 was gradual from the earliest stage of CKD, classical surrogates of UC, such as LVH, remained stable until advanced disease was present. A similar behaviour of native T1 and T2 was later reported across the spectrum of CKD [28]. These findings suggests that T1 and T2 mapping may be used from the very beginning of renal disease to stage and track the adverse changes at the myocardium level.

T1 mapping is a relevant prognostic marker in a variety of cardiac conditions, but outcome data reporting the prognostic value of T1 in the context of CKD are still scarce. A small study that included 52 HD patients showed that, after 38 months of follow-up, native T1 independently predicted major adverse cardiovascular events (MACE) [27]. Additionally, in the specific scenario of severe aortic stenosis and CKD, a native T1 > 1024 ms (1.5T, MOLLI 3(3)5, 35°) was the strongest predictor of MACE after 3.8 years [32]. Although limited by small sample size and other considerations, these studies lead the way for much needed further research that fills the knowledge gap in the prognostic stratification of CKD.

### 3.4. Vascular Stiffness

Observational studies have described increased aortic stiffness, measured as PWV or distensibility, across the spectrum of CKD [15,23,64,65] (Table 2B). Furthermore, distensibility decreases in a staged manner with worsening CKD, and glomerular filtration and age are independently related to distensibility [41]. Although the development of myocardial fibrosis in CKD, measured by native T1, has been shown to happen independently of afterload, probably mediated by mineral bone metabolism and neurohormonal activation among other processes, the increased aortic stiffness reported in CKD patients accelerates this process. In a study with 276 patients, fibrosis and aortic stiffness (expressed as T1 and PWV) had a markedly stronger association in the presence of CKD, suggesting a physiological relationship that is strengthened with the severity of CKD [23].

### 3.5. Other Biomarkers

Other UC features that can be imaged by CMR include microvascular dysfunction by perfusion imaging, which, in one study, was more frequently found in CKD patients than in controls [23], and diastolic function by phase contrast imaging [41], albeit with specific data scarcely available in UC. Feature tracking CMR can be used to derive a parametric function for myocardial deformation, with information comparable to those obtained by speckle-tracking echocardiography [84]. In CKD, reduced longitudinal strain has been found compared to controls [21], with values showing an improvement in ESRD patients after kidney transplantation [47]. Overall, these data are less robust compared with those obtained by tissue characterization and standard cardiac function evaluation.

## 4. Computed Tomography

CT does not represent the first-line test of choice for evaluating cardiovascular involvement in CKD, and imaging biomarkers derived from this modality are not as robust as those obtained by echocardiography and CMR. However, CT of thorax and/or abdomen, performed with other indications, can provide additional ancillary data to support a diagnosis of UC. Myocardial end-diastolic volume and mass can be quantified even with ECG-triggered CT [85]. In patients undergoing coronary CT, the measurement of LV mass and end-diastolic volume, plus its ratio as index of concentric remodelling, was able to differentiate hypertensive from non-hypertensive patients [86], highlighting the potential diagnostic value of this approach. Furthermore, CT-derived LV end-diastolic volume [87] and mass [88] demonstrated prognostic relevance in cohorts of patients undergoing coronary CT. Right ventricular morphology can also be evaluated, with increasing volume being associated with increased mortality in patients with pulmonary embolism [89]. Though no such robust data are available in patients with CKD, previous evidence is likely to be transferrable to this subset of patients, suggesting that CT evaluation of the left and right ventricular chambers can aid risk stratification in this setting.

CKD presents with common and extensive arterial and valvular calcification (Figure 4) due to a pronounced impairment in bone and mineral metabolism, which is easily seen by CT. A disproportionate amount of coronary, aortic and mitral calcium is a well-known finding in patients with ESRD undergoing dialysis [56]. However, even among young dialysis patients (20–30 years old) with otherwise low CV risk, calcification is common and significant; even more importantly, this calcification is rapidly progressive [57]. However, data on coronary artery disease in the earlier stages are more limited. A population study found an association with significant coronary calcification, which was directly related to the stage of renal dysfunction, with no relevant calcification in stages 1–2 compared to a population with no CKD; this association was notably stronger among diabetics [58]. Coronary calcification, quantified by Agatston calcium score (Figure 4A,B), retains prognostic significance in this setting, as outlined by multiple studies [59,60,61]. Calcification of the aortic wall is frequently observed as well. This is associated not only with increased vascular stiffness, as expected, but also with higher degrees of diastolic dysfunction [62], marking a more advanced stage of disease with worse prognosis [63].

The evaluation of myocardial tissue composition by CT is a promising field from which some data are emerging. Extra-cellular volume can be quantified by CT, demonstrating high reproducibility and an age-related increase, which suggest the marker to be consistent with the actual pathologic changes in the myocardium [90]. In patients with amyloidosis, ECV by CT is associated with markers of more advanced disease and higher mortality at follow-up [91]. However, the need for iodinate contrast media administration, which has well-known nephrotoxic effects, especially in patients with underlying pre-existent renal disease [92], limits this application in patients with CKD.

## 5. Conclusions

Multiple imaging modalities contribute to a comprehensive and complementary overview of UC. Echocardiography is a widespread and cheap technique that can be used as a first-line imaging test to assess end-organ damage in CKD. CMR is generally less available than echocardiography; however, it can provide more accurate information to aid an early diagnosis of cardiac involvement in CKD, with its imaging biomarkers more suitable for use as surrogate endpoints in clinical trials testing newer therapeutic approaches. Finally, CT is rarely used to specifically assess UC. Nonetheless, the use of this imaging test is widespread, and much information can be drawn from the ancillary cardiovascular findings obtained during examinations performed with other indications. In summary, a multimodal approach, integrating results from different imaging techniques, can aid a better understanding of the physiopathology of UC and improve the clinical management of patients with CKD.

## Figures and Tables

**Figure 1 biomolecules-13-00773-f001:**
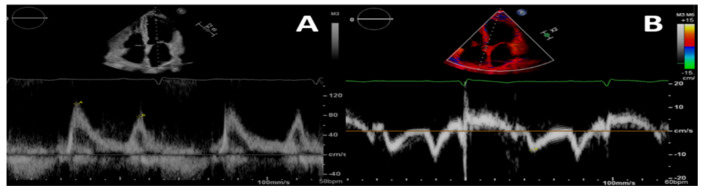
Pulsed-wave Doppler (**A**) and tissue-Doppler (**B**) imaging of the left ventricular basal septum for a 4-chamber apical view in a patient with CKD. This shows a second-degree diastolic dysfunction pattern in A; the E/e’ ratio indicates a likely rise in left ventricular filling pressures.

**Figure 2 biomolecules-13-00773-f002:**
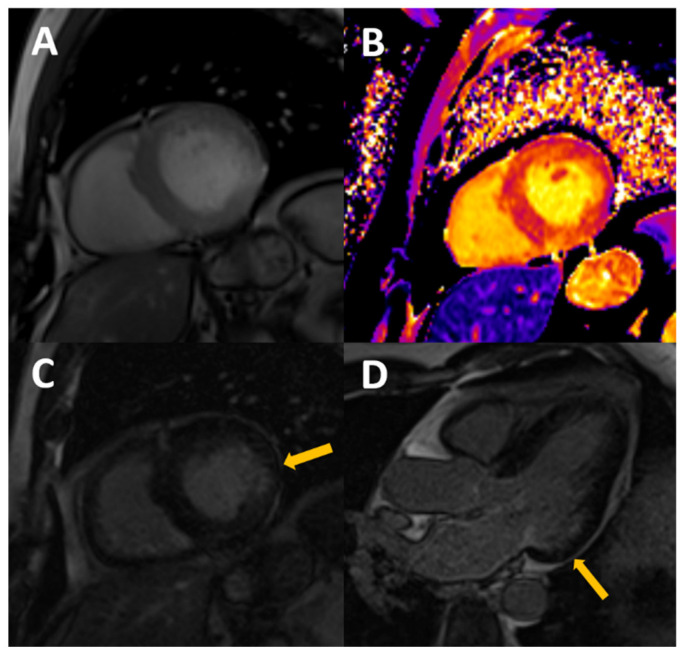
54-year-old male, with stage III CKD secondary to nefroangiosclerosis, who presents with CMR concentric LVH in cine images (**A**), mildly increased native T1 (**B**) with normal T2, probably reflecting appropriate volume status with some degree of diffuse fibrosis. A previously unknown myocardial infarction is present as a subendocardial scar in mid-basal segments of the inferolateral wall (arrows in **C**,**D**).

**Figure 3 biomolecules-13-00773-f003:**
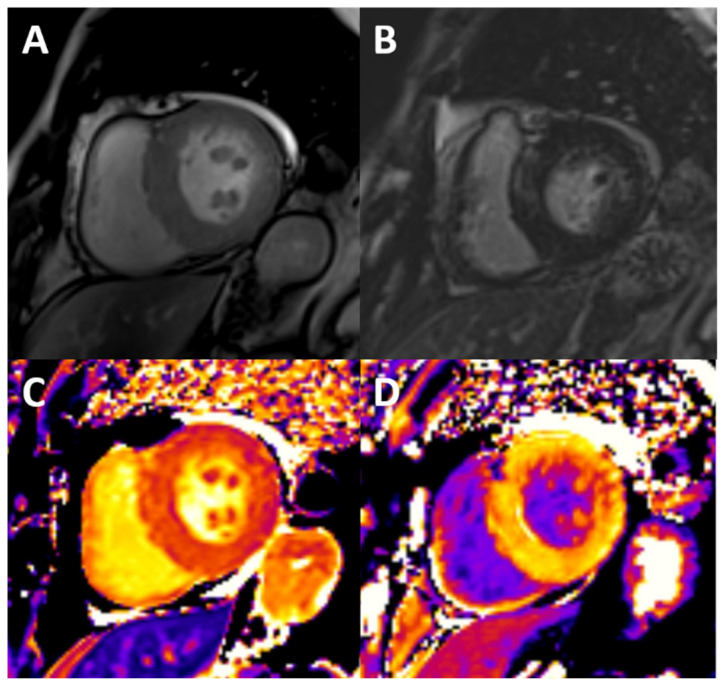
Typical findings of uremic cardiomyopathy with CMR. The patient presents with mild pericardial effusion, severe concentric LVH with hypertrophy of papillary muscles (**A**), diffuse intramyocardial LGE (**B**), and diffuse fibrosis, as shown by high values of native T1 (**C**) and ECV (post contrast T1, **D**).

**Figure 4 biomolecules-13-00773-f004:**
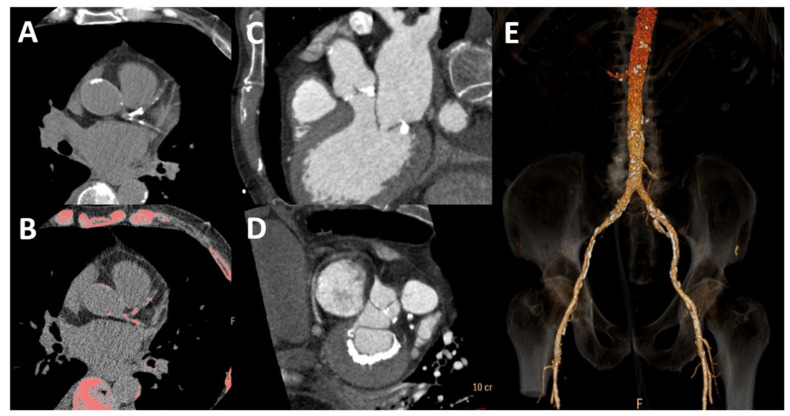
Cardiac and vascular calcification by computed tomography. Coronary calcification is present in bright white (**A**) and pink after post-processing (**B**). There is calcification of the aortic valve (**C**) and posterior mitral anulus (**D**) and vascular calcification of the abdominal aorta and iliac arteries (**E**).

**Table 1 biomolecules-13-00773-t001:** Overview of imaging biomarkers provided by different modalities to assess different cardiac pathologic abnormalities. Legend: - = no data available; + = available data, but not in CKD patients; ++++ = optimal non-invasive biomarker. Prognostic value refers to studies specifically performed in CKD patients. b-SSFP = balanced steady-state free precession; LV = left ventricle; CT = computed tomography; ECV = extra-cellular volume (); LGE = late gadolinium enhancement; PW = pulsed wave; CW = continuous wave.

Pathologic Abnormality	Echocardiography			Cardiac Magnetic Resonance			Computed Tomography			References
	Value	Imaging Biomarker	Prognosis	Value	Imaging Biomarker	Prognosis	Value	Imaging Biomarker	Prognosis	
**LV hypertrophy**	+++	2D- and 3D imaging	Yes	++++	b-SSFP cine imaging	Yes	+	Volumetric CT	No	[8,9,10,11,12,13,14,15,16]
**Fibrosis**	+	Backscatter echocardiography	No	++++	T1 mapping; LGE.	Yes	+	CT ECV	No	[16,17,18,19,20,21,22,23,24,25,26,27,28,29,30,31,32]
**Edema**	-	-	-	++++	T1 and T2 mapping	No	-	-	-	[16,24,25,26,28,29,30]
**Microvascular dysfunction**	++	PW Doppler		++	Perfusion imaging	No	-	-	-	[23,33,34,35]
**LV stiffness**	+++	PW-Doppler; tissue Doppler;sSpeckle tracking	Yes	+	Phase contrast imaging	No	-	-	-	[36,37,38,39,40,41]
**LV systolic dysfunction**	+++	2D-imaging; speckle tracking	Yes	++++	b-SSFP cine imaging; feature tracking	Yes	+	Cine CT	No	[16,21,42,43,44,45,46,47]
**Right heart abnormalities**	+++	2D and 3D imaging; CW-Doppler; tissue Doppler; speckle tracking	Yes	++++	b-SSFP cine imaging; feature tracking	Yes	+	Volumetric CT	No	[36,48,49]
**Calcification**	+++	2D-imaging	Yes	+	T1 and T2 weighted imaging	No	++++	Calcium score	Yes	[50,51,52,53,54,55,56,57,58,59,60,61,62,63]
**Vascular stiffness**	+	Tissue Doppler; speckle tracking	No	+++	Phase-contrast imaging	No	++	Vascular calcification (indirect estimate)	Yes	[15,23,45,64,65]

**Table 2 biomolecules-13-00773-t002:** Summary of studies that reported parameters of diffuse fibrosis (section A) and vascular stiffness (section B) with CMR in different CKD populations compared to controls. Studies are presented in chronological order of publication. Values are reported for 1.5 and 3T in control group (healthy) and CKD group (disease). The last column reports the percentage of patients who presented with LGE, when available, and the proportion of ischaemic aetiology in brackets. Values of T1 and T2 mapping are expressed in ms, PWV in m/s and distensibility in mm Hg^−1^. MOLLI = modified Look-Locker inversion recovery; AA = ascending aorta; PWV = pulse wave velocity.

Author (N)	Population	Sequence	Health	Disease	LGE (%)
1.5T	3T	1.5T	3T
**A. Fibrosis**
**Edwards (43)** [19]	**60–15 mL/min/1.73 m^2^**	Native T1 (MOLLI 3(3)3(3)5)	955 ± 30		986 ± 37		30 (0)
ECV	0.25 ± 0.03		0.28 ± 0.04	
**Graham-Brown (35)** [20]	Hemodialysis	Native T1 (MOLLI 3(3)3(3)5, 50°)		1292.7		1088.8	
**Rutherford (33)** [21]	Hemodialysis	Native T1 (MOLLI 3(3)3(3)5 35°)		1161 ± 29		1184 ± 34	
**Antlanger (37)** [22]	Hemodialysis	Native T1 (MOLLI 5(3)3 35°)	998 ± 47		1022 ± 50		
**Chen (276)**[23]	≤60 mL/min/1.73 m^2^	Native T1 (MOLLI 3(2)3(2)5 50°)		1123 ± 31		1152 ± 43	35 (16)
**Arcari (154)** [24]	≤60 mL/min/1.73 m^2^	Native T1 (MOLLI 3(2)3(2)5 50°)		1062 ± 39		1161 ± 55	7 (4)
T2 FLASH		35.8 ± 2.3		41.8 ± 5.2
**Han (43)** [25]	Hemodialysis	Native T1 (MOLLI 5(3)3 35°)	1006 ± 25		1056 ± 32		-
T2-SSFP	46 ± 2		50 ± 3		
**Lin (23)** [26]	Peritoneal dialysis	Native T1 (MOLLI 5(3)3 35°)		1256 ± 45		1302 ± 30	-
T2-TrueFISP		40.5 ± 1.6		44.6 ± 2.6	
**Qin (52)** [27]	Hemodialysis	Native T1 (MOLLI 5(3)3 20°)		1238 ± 31		1280± 45	
**B. Vascular stiffness**
**Edwards (117)** [64]	60–30 mL/min/1.73 m^2^	AA distensibility	4.12 × 10^−3^		2.94/2.18 × 10^−3^ (stage 3–2)		
**Chue (189)** [65]	90–15 mL/min/1.73 m^2^	AA distensibility	4.1 × 10^−3^		2.8 × 10^−3^		
**Odudu (54)** [15]	Hemodialysis	AA distensibility	4.1 × 10^−3^		2 × 10^−3^		
		PWV	5.3 ± 1.9		7.9 ± 3.5		
**Chen (276)** [23]	≤60 mL/min/1.73 m^2^	PWV		7.3 ± 2.4		9.2 ± 2.6

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
