# Peer review of "Cardiac Imaging Biomarkers in Chronic Kidney Disease"

_biomolecules, 2023, doi:10.3390/biom13050773_

Round 1

Reviewer 1 Report

First of all, thank you for the efforts of the authors to review a bunch of scientific reports about cardiac imaging data in CKD. 

The title needs to revise to understand the readers well. I suggest that the 'cardiac' or 'cardiomyopathy' should be used to represent the article's contents. 

There were lots of data about echocardiography and CMR. However, the author was just referring to these data in line. I suggest that modify the manuscripts into smaller sections to have structure. It will help the readers' understanding. 

Author Response

The title needs to revise to understand the readers well. I suggest that the 'cardiac' or 'cardiomyopathy' should be used to represent the article's contents. 

Title changed accordingly

There were lots of data about echocardiography and CMR. However, the author was just referring to these data in line. I suggest that modify the manuscripts into smaller sections to have structure. It will help the readers' understanding. 

The text has been divided into smaller sections for an easier Reading. Thank you for your suggestion

Reviewer 2 Report

This concise review article by Arcari et al, attempts to summarize pros and cons of various imaging methods, echo, CMR, CT, in assessing uremic cardiomyopathy.  Conclusions drawn are that multimodality approach should be taken to improve diagnosis and management of patients with CKD.  In that regard, one can view the necessity of this review is low as it does not add much value to clinicians in their selection of imaging modalities, which is limited by real life issues of clinical infrastructure and patient accessibility.  

Minor comments:

1) CMR paragraph is too long with too many rambling sentences. 

2) T1, T2 needs to be defined so that readers do not have to find the referenced article. 

Major comments:

1) Table legends are incomplete. Both lack a title that describes the purpose of the tables.  The description of table columns are lacking.  Where did the information come from? The papers should be referenced within the column.

2) Table 2 legend doesn't describe A or B.  Also the units for the scores shown under health and disease are missing.  The diffuse fibrosis mentioned does not correlate with values shown from selected publications. Selected publications need reference numbers, in addition to author name in column 1. Table 2 does not appear to show any T2, but it is mentioned in legend. 

3) Table 1 needs better description.  Where did value and prognosis score come from?  The writer's opinion or actual published reports?  Table needs a reference column where articles used (if any) are listed.

Author Response

This concise review article by Arcari et al, attempts to summarize pros and cons of various imaging methods, echo, CMR, CT, in assessing uremic cardiomyopathy.  Conclusions drawn are that multimodality approach should be taken to improve diagnosis and management of patients with CKD.  In that regard, one can view the necessity of this review is low as it does not add much value to clinicians in their selection of imaging modalities, which is limited by real life issues of clinical infrastructure and patient accessibility.  

The number of CKD patients is rising and these patients frequently undergo cardiac exams (mainly echo, but also CMR and CT). This review tries to summarize which parameters are important to diagnosis and prognosis and how to interpret them, facilitating interpretation of cardiac examinations that are anyway routinely performed. CKD population has long been neglected in clinical trials but it has received more attention lately in interventional trials and also new therapies that improve outcomes (i.e. iSGLT2), therefore knowing how to measure the efficacy of these medical interventions from a CV point of view is important.

Minor comments:

1) CMR paragraph is too long with too many rambling sentences. 

The text has been divided into sections and this paragraph simplified.

2) T1, T2 needs to be defined so that readers do not have to find the referenced article. 

Practical interpretation of T1 and T2 are depicted in the text

Major comments:

1) Table legends are incomplete. Both lack a title that describes the purpose of the tables.  The description of table columns are lacking.  Where did the information come from? The papers should be referenced within the column.

Legends have been reviewed and titles completed, as well as columns explanation. References are in each row.

2) Table 2 legend doesn't describe A or B.  Also the units for the scores shown under health and disease are missing.  The diffuse fibrosis mentioned does not correlate with values shown from selected publications. Selected publications need reference numbers, in addition to author name in column 1. Table 2 does not appear to show any T2, but it is mentioned in legend. 

A and b explained in the accompanying text. Tge units are also reported there. Missing references have been completed. T2 values are reported in Arcari, Han and Lin.

3) Table 1 needs better description.  Where did value and prognosis score come from?  The writer's opinion or actual published reports?  Table needs a reference column where articles used (if any) are listed.

Completed in the legend. The table is complementary to  the text and tries to simplify and summarize the information from the text (quick overview of the biomarkers with evidence that support their use), it does not correspond to personal opinion, but to published reports, the references have been added in an additional column.

Reviewer 3 Report

The paper is well written and provided an important message for the clinical field, specifically, to add the cardiovascular imaging biomarkers for the characterization and risk-stratification of uremic cardiopathy. 

I do not have additional comments to the Authors. 

Author Response

English style reviewed

Reviewer 4 Report

Review  of the manuscript :  Imaging biomarkers in chronic kidney disease.

The aim of manuscript was  review  studies on the role of CV imaging in the assessment of CKD-related myocardial remodelling, with a focus on echocardiography, cardiac magnetic resonance (CMR) imaging and  computed tomography (CT).

In my opinion  review is interesting and  my be useful in  clinical practice and also in  future  planned researches. 

I have no comments.

Author Response

Thank you

Round 2

Reviewer 2 Report

No further comments. Tables are now complete and editorial changes suggested have been included.